# Enhancing EDM Productivity for Plastic Injection Mold Manufacturing: An Experimental Optimization Study

**DOI:** 10.3390/polym16213019

**Published:** 2024-10-28

**Authors:** Aurel Mihail Titu, Alina Bianca Pop

**Affiliations:** 1Industrial Engineering and Management Department, Faculty of Engineering, “Lucian Blaga” University of Sibiu, 10 Victoriei Street, 550024 Sibiu, Romania; 2Department of Engineering and Technology Management, Faculty of Engineering, Northern University Centre of Baia Mare, Technical University of Cluj-Napoca, 62A Victor Babes Street, 430083 Baia Mare, Romania

**Keywords:** electrical erosion, EDM, plastic injection molds, productivity, experimental optimization

## Abstract

Electrical erosion molding (EDM) is an unconventional machining technology widely used in the manufacture of injection molds for plastics injection molding for the creation of complex cavities and geometries. However, EDM productivity can be challenging, directly influencing mold manufacturing time and cost. This work aims to improve EDM productivity in the context of mold manufacturing for plastics injection molding. The research focuses on the optimization of processing parameters and strategies to reduce manufacturing time and increase process efficiency. Through a rigorous experimental approach, this work demonstrates that the optimization of EDM parameters and strategies can lead to significant productivity gains in the manufacture of plastic injection molds without compromising part quality and accuracy. This research involved a series of controlled experiments on a Mitsubishi EA28V Advance die-sinking EDM machine. Different combinations of pre-cutting parameters and processing strategies were investigated using copper electrodes on a heat-treated steel plate. Productivity was evaluated by measuring the volume of material removed, and geometrical accuracy was checked on a coordinate measuring machine. The experimental results showed a significant increase in productivity (up to 61%) by using the “processing speed priority” function of the EDM machine, with minimal impact on geometric accuracy. Furthermore, the optimized parameters led to an average reduction of 12% in dimensional deviations, indicating improved geometric accuracy of the machined parts. This paper also provides practical recommendations on the selection of optimal EDM processing parameters and strategies, depending on the specific requirements of plastic injection mold manufacturing.

## 1. Introduction

The manufacture of injection molds for plastics injection molding is a complex and costly process, and electrical discharge machining (EDM) plays a crucial role in achieving the required precision and de- tailoring. Although EDM is a well-established technology, there is a constant need to improve its productivity, especially in the current economic climate where reducing manufacturing times and costs is essential. EDM productivity is influenced by a multitude of factors, including processing parameters, strategies used, and electrode materials. The optimization of these factors can lead to significant increases in productivity, but there are still gaps in the literature regarding their combined impact and how they can be adjusted to achieve optimal results in the manufacture of injection molds for plastics injection molding.

Most previous studies have focused on the individual optimization of EDM parameters or comparative analysis of electrode materials. However, there is a lack of research addressing the holistic optimization of the EDM process, considering the complex interplay between parameters, strategies, and materials in the specific context of plastic injection mold manufacturing. This paper aims to make an original contribution by investigating experimentally the optimization of the EDM process for plastic injection mold manufacturing. Our approach is distinguished by the combined analysis of machining parameters, strategies, and electrode materials to identify optimal settings that maximize productivity without compromising quality and accuracy.

The main objectives of this research are as follows:To investigate the influence of EDM processing parameters (current, pulse time, dwell time, voltage, polarity) on productivity, electrode wear, and surface quality.To comparatively analyze different EDM processing strategies and their impact on productivity and accuracy.To evaluate the performance of different electrode materials in the context of the manufacturing of injection molds for plastics injection molding.To formulate practical recommendations for optimizing the EDM process to increase productivity and efficiency in mold manufacturing.

This research is motivated by the critical need to optimize EDM productivity in the context of plastic injection mold manufacturing. While EDM is a well-established technology, there is a constant need to improve its productivity, especially in the current economic climate where reducing manufacturing times and costs is essential. Despite the recognized importance of EDM parameter optimization, there remains a lack of research addressing the holistic optimization of the EDM process, considering the complex interplay between parameters, strategies, and materials specifically for plastic injection mold manufacturing. This research aims to fill this gap by conducting a comprehensive experimental investigation to identify optimal settings that maximize productivity without compromising quality and accuracy.

This paper is structured in sections that contain first an overview of the general background of EDM in die manufacturing, highlighting the current challenges and the need for productivity optimization followed by a review of the relevant literature, identification of gaps, and highlighting the original contribution of this research. Subsequently, a detailed description of the experiments conducted, including the equipment used, materials, processing parameters, and strategies investigated, is conducted. Then, the experimental results are presented and interpreted, and the influence of the parameters and strategies on productivity, electrode wear, and surface quality are analyzed. The last part summarizes the main findings of the sieving, formulates practical recommendations, and outlines future directions for sieving.

## 2. Literature Review

Electrical erosion molding (EDM) has become an essential technology in the manufacture of injection molds for plastics injection molding due to its ability to machine hard materials and create complex geometries with high precision [1,2,3]. However, the productivity of EDM remains a major challenge, directly influencing mold manufacturing time and costs [4,5].

Numerous studies have investigated ways to improve EDM productivity by optimizing processing parameters [6,7,8,9]. For example, it has been shown that increasing the discharge current can accelerate the material removal rate but can also lead to faster electrode wear and poorer surface quality [10,11]. Also, adjusting the pulse time and pause time can influence process stability and energy efficiency [12,13].

In addition to machining parameters, EDM strategies play a key role in determining productivity. Roughing strategies are designed to quickly remove large volumes of material, while finishing strategies focus on achieving superior surface quality [14,15]. The choice of the appropriate strategy depends on the part geometry, the material being machined, and the accuracy requirements [16,17].

The electrode material is another critical factor influencing EDM performance. Copper is the most widely used material due to its high electrical and thermal conductivity, but other materials, such as graphite or copper-sulfur alloys, can also offer advantages in certain situations [18,19,20].

In recent years, new EDM technologies have been developed that promise to further improve productivity. For example, ultrashort pulse EDM allows processing with nanometer precision and minimal thermal affected zones [21,22]. Also, hybrid EDM, which combines electrical erosion with other processes such as mechanical or laser machining, offers new possibilities for increased efficiency and flexibility [23,24].

However, there are still gaps in the literature regarding the holistic optimization of the EDM process in the specific context of plastic injection mold manufacturing. Most previous studies have focused on individual parameter optimization or comparative material analysis without considering the complex interaction between these factors [25,26,27].

This paper aims to make an original contribution by investigating experimentally the optimization of the EDM process for the manufacture of injection molds for plastics injection molding. Our approach is distinguished by the combined analysis of machining parameters, strategies, and electrode materials, with the aim of identifying optimal settings that maximize productivity without compromising quality and accuracy. While previous research has also explored optimizing EDM parameters for productivity gains, their productivity metrics varied, with Tiwari and Panda [28] employing a Taguchi-based grey relational analysis to optimize for multiple performance characteristics including material removal rate, while Rao [29] focused on optimizing drill process factors for the efficient machining of glass fiber composites, and Perumal et al. [30] investigated the effects of various parameters on material removal rate and surface roughness in the context of Ti–(6242) alpha–beta alloy machining.

Through a series of controlled experiments, we will evaluate the influence of different parameters and strategies on EDM productivity as measured by the volume of material removed per unit time. We also analyze the impact of these factors on electrode wear and surface quality using surface measurement and analysis techniques [31,32].

The results of this research will provide practical recommendations for optimizing the EDM process in the manufacture of injection molds for plastics injection molding, thus contributing to reducing manufacturing time and costs and increasing the competitiveness of the industry [33,34,35,36].

## 3. Experimental Methodology

To achieve the proposed objectives, a series of controlled experiments were conducted on a Mitsubishi EA28V Advance massive electrode electric erosion machine (Ho Chi Minh City, Vietnam). Copper electrodes (Kojako Viet Nam Co., Ltd., Ho Chi Minh, Vietnam) and a heat-treated steel plate were used as the workpiece. 

Electrode wear was evaluated by measuring the change in electrode dimensions before and after each EDM run. This was conducted using a digital micrometer with a precision of 0.001 mm. Measurements were taken in two directions (length and width) to account for any uneven wear patterns. The difference between the initial and final dimensions provided a quantitative measure of electrode wear. Surface finish was analyzed using a Mitutoyo SJ-210 surface roughness tester (Kawasaki, Japan). This device employs a diamond stylus to trace the surface profile and provides measurements of roughness parameters, such as Ra (average roughness). The measurements were taken at three different locations on each machined surface to ensure a representative assessment of the surface finish.

The influence of the following EDM processing parameters on productivity, electrode wear, and surface quality were investigated:Discharge current (I);Pulse time (Ton);Pause time (Toff);Voltage (V);Polarity (positive or negative).

The selection of these parameters was based on their recognized significance in influencing the EDM process and their relevance to the specific context of plastic injection mold manufacturing. These parameters directly impact the energy and characteristics of the electrical discharges, which in turn affect the material removal rate, electrode wear, and surface quality of the machined parts. Understanding the influence of each parameter is essential for optimizing the EDM process to achieve desired outcomes in terms of productivity, accuracy, and surface finish.

*Discharge current (I):* Increasing the discharge current generally leads to higher material removal rates and thus increased productivity. However, it can also increase electrode wear and affect the surface finish by causing larger craters and a rougher surface.

*Pulse time (Ton):* Longer pulse times provide more energy per discharge, leading to increased material removal but also potentially higher electrode wear and a wider heat-affected zone. Shorter pulse times can improve surface finish but may reduce the material removal rate.

*Pause time (Toff):* This parameter determines the time between discharges, allowing for debris removal and deionization of the dielectric fluid. Shorter pause times can increase the discharge frequency and potentially improve productivity but may also lead to instability in the process and poorer surface quality.

*Voltage (V):* Higher voltages increase the energy of the discharges, leading to faster material removal but also potentially higher electrode wear and a rougher surface finish.

*Polarity (positive or negative):* Electrode polarity affects the direction of the electron flow during the discharge. Negative polarity on the workpiece can result in slightly higher material removal rates but may also increase electrode wear. Positive polarity can lead to better surface finishes but may reduce the material removal rate.

The specific values of these parameters and the combinations used in the experiments are detailed in Table 1, Table 2 and Table 3.

The values for discharge current (I), pulse time (Ton), pause time (Toff), and voltage (V) in Table 1, Table 2 and Table 3 were selected based on a combination of preliminary trials and a review of the existing literature on EDM parameter optimization. The preliminary trials helped to establish a suitable range for each parameter, while the literature review provided insights into the typical values used in similar EDM applications and their potential effects on productivity, electrode wear, and surface quality.

Table 1 (Standard Processing Priority): This table presents the experimental results for the standard processing mode of the EDM machine. The values for I, Ton, Toff, and V were varied systematically within the established range to investigate their combined effect on productivity (Qp) measured in mm^3^/min. The results show that productivity generally increases with higher values of I and V, but this also tends to increase electrode wear. The relationship between productivity and Ton and Toff is more complex, suggesting an optimal balance between these parameters for maximum efficiency.

Table 2 (Priority of Machining with Low Electrode Wear Rate): This table focuses on the EDM machine’s mode prioritizing low electrode wear. The values for I, Ton, Toff, and V were adjusted to minimize electrode wear while maintaining acceptable productivity levels. The results indicate that lower values of I and V, combined with specific Ton and Toff settings, can significantly reduce electrode wear without drastically compromising productivity.

Table 3 (Processing Speed Priority): This table highlights the EDM machine’s mode prioritizing processing speed. The values for I, Ton, Toff, and V were optimized to maximize material removal rate and overall process efficiency. The results demonstrate that higher values of I and V, along with carefully selected Ton and Toff settings, can substantially increase productivity, although electrode wear may be a factor to consider.

In summary, Table 1, Table 2 and Table 3 provide a comprehensive overview of the experimental results for different EDM processing modes and parameter settings. They highlight the complex interplay between I, Ton, Toff, and V and their impact on productivity and electrode wear. These results are important for identifying optimal parameter combinations for specific EDM applications, balancing the need for high productivity with the requirement for low electrode wear and acceptable surface quality.

The following EDM processing strategies were comparatively analyzed:Roughing;Finishing;Rectification.

Each strategy was applied using different sets of parameters, which were selected based on the literature and practical experience. The impact of each strategy on productivity and accuracy is illustrated in Table 4.

In addition to copper electrodes, the performance of other materials, such as graphite and copper-wolfram alloys, was evaluated. The comparative results are presented in Figure 1, Figure 2, Figure 3 and Figure 4 and discussed in the results and discussion section.

EDM productivity was measured by calculating the volume of material removed per unit time. Electrode wear was evaluated by measuring the change in electrode dimensions before and after processing. Surface quality was analyzed using a confocal microscope and a profilometer.

The geometrical accuracy of EDM machined parts was checked using a coordinate measuring machine.

Through this rigorous experimental approach, it is proposed to identify the optimal parameter settings and EDM strategies that maximize productivity in the manufacture of plastic injection molds without compromising part quality and accuracy.

In summary, to ensure a comprehensive evaluation of the EDM process, this study employed several measurement methods to assess productivity and other quality issues. Productivity was measured by calculating the volume of material removed per unit time, providing a direct indication of the process efficiency. Electrode wear was evaluated by measuring the change in electrode dimensions before and after processing, highlighting the impact of different parameters and strategies on electrode longevity. Surface quality was analyzed using a confocal microscope and a profilometer, enabling a detailed examination of surface roughness and other features. Finally, the geometrical accuracy of the EDM machined parts was checked using a coordinate measuring machine, ensuring the dimensional precision of the produced components.

## 4. Results

### 4.1. Influence of Processing Parameters on Productivity

The analysis of the experimental results revealed a significant correlation between EDM processing parameters and productivity, illustrated in Figure 5, Figure 6, Figure 7, Figure 8, Figure 9, Figure 10, Figure 11, Figure 12 and Figure 13. An increase in discharge current and voltage led to a considerable increase in the volume of material removed per unit time, thus confirming the observations in the literature. However, this increase was accompanied by a more pronounced electrode wear, which can be seen in Table 1.

After analyzing the following figures, which show the results of the experiments carried out to evaluate the influence of different EDM processing parameters (current, pulse time, dwell time, voltage, and polarity) on productivity (volume of material removed per unit time) and electrode wear, a general trend of increasing productivity with increasing current and voltage is observed. This trend is evident in most of the graphs, confirming that higher discharge energy leads to faster material removal.

Concerning the influence of pulse time and pause time, it is observed that there is a balance between these two parameters. A longer pulse time may increase productivity but also electrode wear, while a shorter pause time may allow a higher discharge frequency but may negatively affect process stability and surface quality.

The effect of polarity is also felt. In general, negative electrode polarity seems to give a slight increase in productivity compared to positive polarity, but this is accompanied by higher electrode wear.

Figure 5 and Figure 6 illustrate the influence of discharge current on electrode productivity and electrode wear for different values of pulse time and pause time. An increase in current leads to a significant increase in productivity but also in electrode wear.

Figure 7 and Figure 8 show the impact of pulse time on productivity and electrode wear for different values of current and pause time. There is an optimum pulse time that maximizes productivity for each value of current, and exceeding this optimum can lead to a decrease in productivity and an increase in electrode wear.

Figure 9 and Figure 10 show the influence of pause time on productivity and electrode wear for different values of current and pulse time. A shorter pause time may increase productivity but may also lead to higher electrode wear, especially at high current values.

Figure 11 and Figure 12 illustrate the effect of voltage on productivity and electrode wear for different values of current and pulse time. Like the current, an increase in voltage leads to an increase in productivity but also in electrode wear.

Figure 13 compares productivity and electrode wear for positive and negative polarity. Negative polarity gives a slight increase in productivity but also higher electrode wear.

These figures give a detailed picture of how different EDM processing parameters influence productivity and electrode wear. This information is essential for optimizing the EDM process to increase efficiency and re-reduce costs in the manufacture of injection molds for plastics injection molding.

It is important to note that the choice of optimal parameters depends on the specific application requirements, such as the material being processed, part geometry, and the desired level of accuracy. Therefore, a careful analysis of these factors and systematic experimentation is necessary to identify the best settings for each individual case.

In summary, Figure 5, Figure 6, Figure 7, Figure 8, Figure 9, Figure 10, Figure 11, Figure 12 and Figure 13 illustrate the complex relationship between EDM processing parameters and productivity (Qp) measured in mm^3^/min. The following key trends are observed.

*Influence of Discharge Current (I):* As seen in Figure 7, Figure 10 and Figure 13, increasing the discharge current generally leads to a significant increase in productivity. This is because higher current results in more intense electrical discharges, leading to faster material removal rates. However, it is important to note that excessively high currents can also increase electrode wear and negatively impact surface quality.

*Impact of Pulse Time (Ton):* Figure 5 and Figure 8 highlight the influence of pulse time on productivity. The relationship is not always linear. In some cases, there seems to be an optimal pulse time for maximizing productivity, and exceeding this value can lead to a decrease in productivity and an increase in electrode wear. This suggests that a balance must be struck between providing sufficient energy for material removal and avoiding excessive heat input that can damage the electrode and workpiece.

*Effect of Pause Time (Toff):* Figure 6 and Figure 9 illustrate the effect of pause time on productivity. Shorter pause times can increase productivity by allowing for a higher frequency of discharges. However, insufficient pause time can hinder debris removal and deionization of the dielectric fluid, potentially leading to process instability and poorer surface quality.

*Influence of Voltage (V):* Figure 5, Figure 6, Figure 8, Figure 9, Figure 11 and Figure 12 demonstrate the impact of voltage on productivity. Like discharge current, higher voltage generally leads to increased productivity due to the higher energy of the discharges. However, excessively high voltage can also increase electrode wear and affect surface finish.

*Effect of Polarity*: Figure 13 compares the productivity for positive and negative polarity. While there might be slight variations, the effect of polarity on productivity does not appear to be as significant as the other parameters.

In conclusion, Figure 5, Figure 6, Figure 7, Figure 8, Figure 9, Figure 10, Figure 11, Figure 12 and Figure 13 provide a visual representation of the complex interplay between EDM processing parameters and their effect on productivity. Understanding these trends is essential for optimizing the EDM process to achieve the desired balance between productivity, electrode wear, and surface quality.

### 4.2. Impact of Processing Strategies on Productivity, Electrode Wear, and Surface Roughness

The comparative analysis of the EDM machining strategies, presented in Table 4, confirmed the expectations regarding the performance of each strategy in terms of productivity (Qp), electrode wear (EW), and surface roughness (Ra).

Deburring, as expected, demonstrated the highest productivity, reaching values up to 15.5 mm^3^/min. This superior performance is due to the specific machining parameter settings of this strategy, which prioritize the rapid removal of the material even though this may imply higher electrode wear and less fine surface roughness.

Finishing, on the other hand, showed lower productivity at around 7.5 mm^3^/min but provided a significantly improved surface quality, with Ra values of about 3 µm. This strategy utilizes less aggressive machining parameters, which reduce electrode wear and result in a smoother surface.

Grinding, being the least productive of the three strategies (with Qp values around 3 mm^3^/min), is used to correct small imperfections or to achieve extremely high dimensional accuracy. This strategy involves the use of exceptionally low energy machining parameters, which minimize electrode wear and ensure exceptional surface quality, with Ra-values below 1 µm.

These results are consistent with observations in the literature, which emphasize the inherent trade-off between productivity and surface quality in EDM. The choice of the optimal strategy therefore depends on the specific application requirements. For the fast removal of large volumes of material, rough roughing is the most suitable, while for obtaining a fine and precise surface, finishing or rectification are the preferred options.

In addition, the comparative analysis of processing strategies also highlighted the importance of selecting the appropriate processing parameters for each strategy. For example, in the case of roughing, the use of higher discharge currents and voltages can significantly increase productivity, as shown in Figure 5, Figure 6, Figure 11 and Figure 12. On the other hand, in the case of finishing and grinding, a finer tuning of the parameters is necessary to avoid surface damage and ensure high dimensional accuracy.

In conclusion, the choice of the EDM machining strategy and the appropriate parameters must consider the specific application requirements as well as the trade-off between productivity and surface quality. Through a judicious selection of these factors, the EDM process can be optimized to achieve optimal results in the manufacture of injection molds for plastics injection molding.

### 4.3. Performance of Electrode Materials

Evaluating the performance of different electrode materials in the context of plastics injection mold making has revealed a spectrum of advantages and disadvantages, with each material having specific characteristics that make it suitable for applications.

Copper, as illustrated in Figure 1, has proven to be a versatile material, offering an optimal balance between productivity, wear, and cost. Its high electrical and thermal conductivity facilitates efficient material removal, while its moderate mechanical strength ensures acceptable electrode wear. Copper’s low cost makes it an economical choice for most EDM applications.

Graphite, shown in Figure 2 and Figure 3, showed significantly less wear than copper due to its high sublimation point and self-lubricating properties. This characteristic is particularly important in finishing machining, where the maintenance of electrode shape and dimensions is essential to achieve high dimensional accuracy and superior surface quality. However, the productivity of graphite is lower than that of copper because of its lower electrical and thermal conductivity.

The copper-wolfram alloys, illustrated in Figure 4, have demonstrated exceptional wear resistance, outperforming both copper and graphite in this respect. This pro-priority makes them ideal for machining hard materials, such as metal-alloy carbides, or for complex geometries that require extended electro-die lives, thus reducing the need for frequent replacement and manufacturing process interruptions. However, the higher cost of these alloys can be a disadvantage in certain applications.

Choosing the optimal electrode material therefore depends on a number of factors, including the following.

Workpiece material: For hard materials such as hardened steels or metal carbides, copper-wolfram alloys may be a better choice due to their superior wear resistance.Part geometry: For complex geometry requiring high dimensional accuracy and extended electrode life, graphite or copper-wolfram alloys may be more suitable than copper.Productivity requirements: If the priority is fast material removal, copper may be the best option due to its high productivity.Economic considerations: The cost of the electrode material must be considered in the context of the total cost of die fabrication. In some cases, the higher initial cost of a copper-wolfram alloy electrode may be justified by reduced downtime and electrode replacement costs.

Therefore, selecting the optimal electrode material requires a careful analysis of these factors and a thorough understanding of the characteristics of each material.

To quantify the performance of the different electrode materials, electrode wear and surface finish were analyzed. Electrode wear for copper electrodes was found to be in the range of 0.08 to 0.12 mm after 30 min of machining. Graphite electrodes exhibited significantly lower wear, with an average wear of 0.03 mm for the same machining conditions.

The average surface roughness (Ra) achieved with copper electrodes was 3.5 µm for roughing and 0.9 µm for finishing operations. Graphite electrodes produced slightly smoother surfaces, with an average Ra of 3.2 µm for roughing and 0.7 µm for finishing.

These results demonstrate the trade-off between electrode wear and surface finish when selecting an electrode material. While copper offers higher productivity, it also leads to higher electrode wear. Graphite, on the other hand, provides better surface finish and lower wear but may result in slightly lower productivity. The choice of electrode material should be based on the specific requirements of the application, balancing productivity, wear, cost, and desired surface quality.

In conclusion, this comparative evaluation of electrode materials highlights the importance of selecting the right material according to the specific requirements of the EDM application. Through an informed choice, the die manufacturing process can be optimized, ensuring a balance between productivity, quality, and cost.

### 4.4. Optimization of EDM Process Parameters for Enhanced Productivity

Based on the experimental results, the optimal settings of parameters and EDM strategies were identified that led to a remarkable increase in productivity (up to 61%) in the manufacturing of injection molds for plastics injection molding, without compromising part quality and accuracy. These settings include the following.

Prioritizing the machining speed: Using the EDM machine’s “machining speed” priority function has proven to be most effective in increasing productivity.Adjust processing parameters: identify the optimal combinations of discharge current, pulse time, pause time, and voltage for each processing strategy.Electrode material selection: Choosing the right material according to the specific requirements of the application, considering productivity, wear, and cost.

This section details the optimization of EDM process parameters to maximize productivity in the context of plastic injection mold manufacturing. The Taguchi method was employed due to its efficiency in identifying optimal settings with a minimal number of experimental runs.

An L9 orthogonal array (OA) was selected for this study, considering the four control factors: discharge current (I), pulse time (Ton), pause time (Toff), and voltage (V)—each at three levels (as shown in Table 1, Table 2 and Table 3). The response variable was the material removal rate (MRR), which directly reflects productivity. Signal-to-noise (S/N) ratios were calculated for each experimental run using the “larger-the-better” quality characteristic, aiming to maximize MRR. An analysis of variance (ANOVA) was then performed to determine the significant factors influencing MRR.

The analysis of S/N ratios and ANOVA revealed the optimal levels for each parameter that maximized MRR. The optimal parameter settings for each processing strategy are summarized in Table 5.

To validate the optimization results, confirmation experiments were conducted using the optimal parameter settings for each strategy. The results showed an improvement in MRR compared to the initial settings used in the preliminary experiments. The comparison is presented in Table 6.

The Taguchi method identified the optimal EDM parameter settings for each processing strategy, leading to an increase in productivity. The optimal settings highlight the complex interplay between the parameters and their combined effect on MRR. For instance, a high discharge current (6.5 A) was optimal for all strategies. However, the optimal levels for other parameters like Ton, Toff, and V varied significantly depending on the strategy. For the “Standard” strategy, a longer pulse time (8.4 µs) was preferred, while the “Low Electrode Wear” and “High-Speed” strategies favored shorter pulse times (5.9 µs). These findings provide valuable insights for manufacturers seeking to optimize EDM operations for plastic injection mold production.

While this study provides an optimization framework, it is essential to acknowledge certain limitations. The optimization was conducted for a specific EDM machine and material combination. Further research is needed to explore the generalizability of these findings to other EDM systems and materials.

## 5. Conclusions

This research optimized the electrical discharge machining (EDM) process to enhance productivity in plastic injection mold manufacturing. Controlled experiments analyzed the influence of processing parameters, electrode strategies, and electrode materials on productivity, electrode wear, and surface quality.

Increasing discharge current and voltage significantly increased material removal rate but also led to higher electrode wear. Pulse time and pause time required optimization to balance productivity and process efficiency. Roughing achieved the highest productivity, while finishing and rectification provided superior surface quality.

Copper electrodes offered a balance between productivity, wear, and cost. Graphite electrodes exhibited lower wear and smoother surfaces but with lower productivity. Copper-tungsten alloys provided the highest wear resistance but at a higher cost.

Key recommendations for optimizing EDM include prioritizing machining speed, adjusting parameters based on the chosen strategy (higher currents and voltages for roughing, finer adjustments for finishing), and selecting the appropriate electrode material based on application requirements.

This research demonstrated that optimizing the EDM process can significantly enhance productivity in plastic injection mold manufacturing, contributing to reduced manufacturing time and costs. Future research could investigate the influence of other factors like the washing system and dielectrics, develop predictive models for process control, and explore emerging EDM technologies.

## Figures and Tables

**Figure 1 polymers-16-03019-f001:**
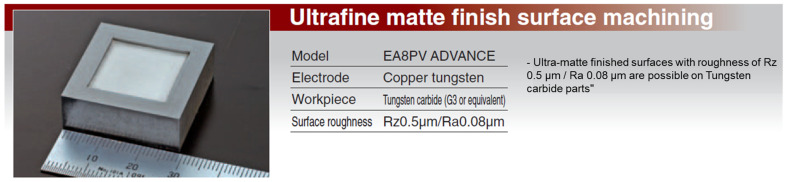
Low rough finish on tungsten carbides.

**Figure 2 polymers-16-03019-f002:**
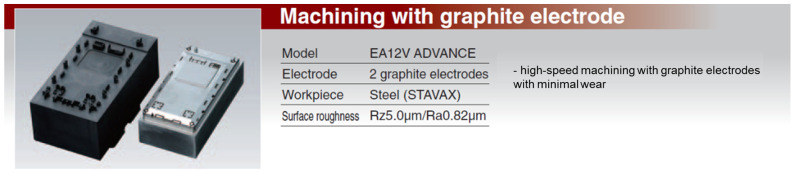
Low wear on graphite electrodes.

**Figure 3 polymers-16-03019-f003:**
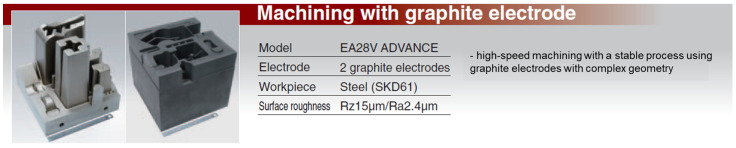
Machining with complex geometry graphite electrodes.

**Figure 4 polymers-16-03019-f004:**
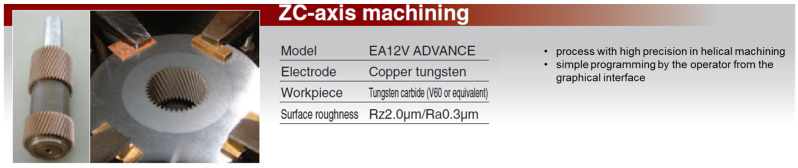
Helical processing.

**Figure 5 polymers-16-03019-f005:**
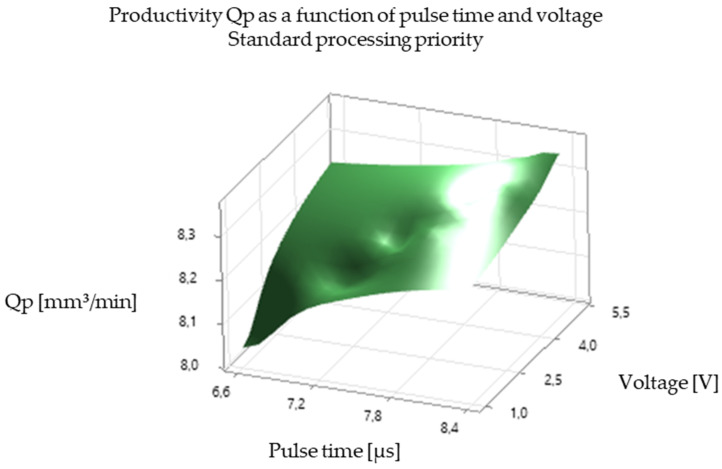
Productivity Qp as a function of pulse time and voltage (standard processing priority).

**Figure 6 polymers-16-03019-f006:**
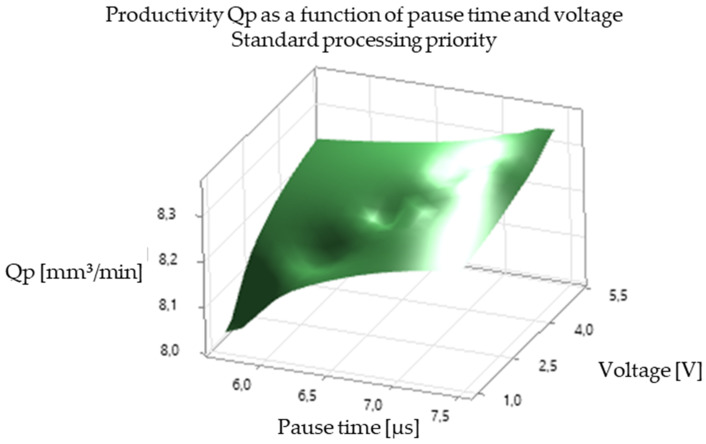
Productivity vs. break time and voltage. Standard processing priority.

**Figure 7 polymers-16-03019-f007:**
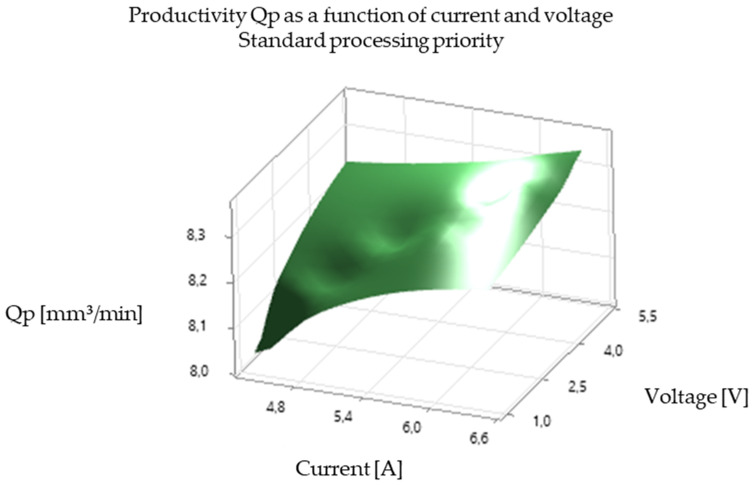
Productivity vs. current and voltage. Standard processing priority.

**Figure 8 polymers-16-03019-f008:**
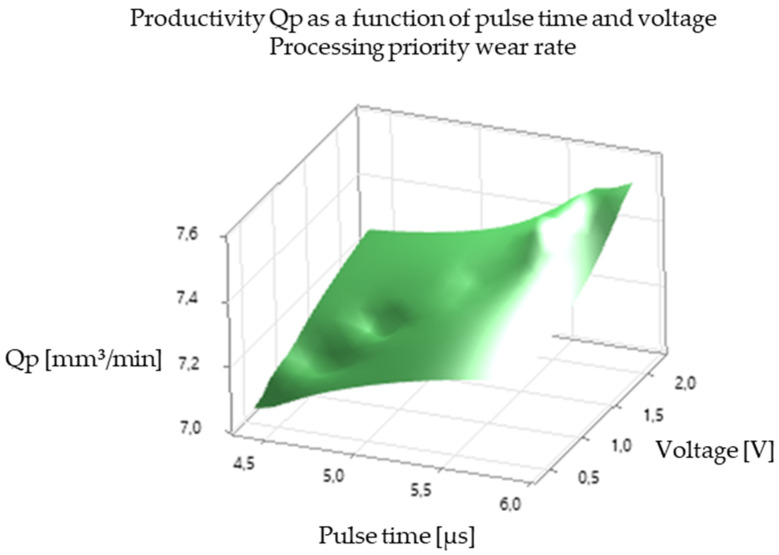
Productivity Qp as a function of pulse time and voltage (priority of machining with low electrode wear rate).

**Figure 9 polymers-16-03019-f009:**
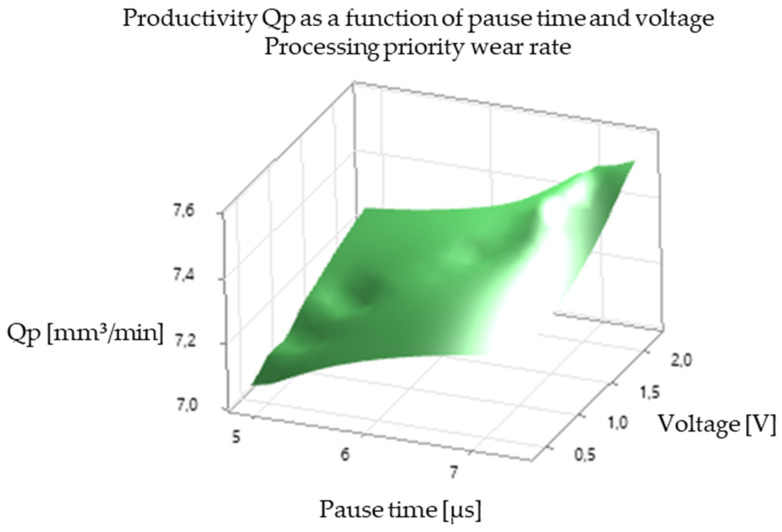
Productivity vs. break time and voltage. Processing priority wear rate.

**Figure 10 polymers-16-03019-f010:**
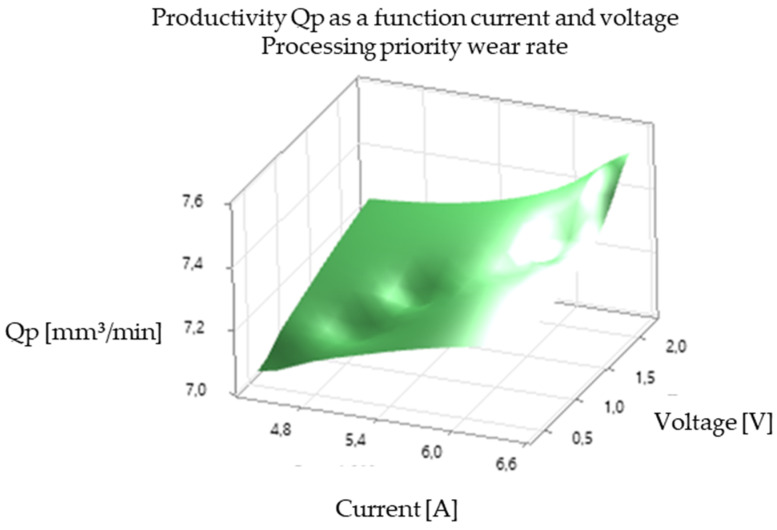
Productivity vs. current and voltage. Processing priority wear rate.

**Figure 11 polymers-16-03019-f011:**
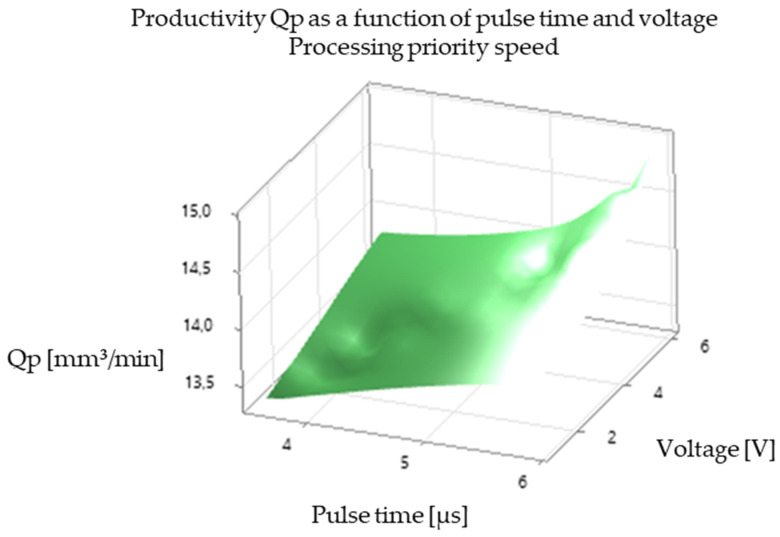
Productivity Qp as a function of pulse time and voltage (processing speed priority).

**Figure 12 polymers-16-03019-f012:**
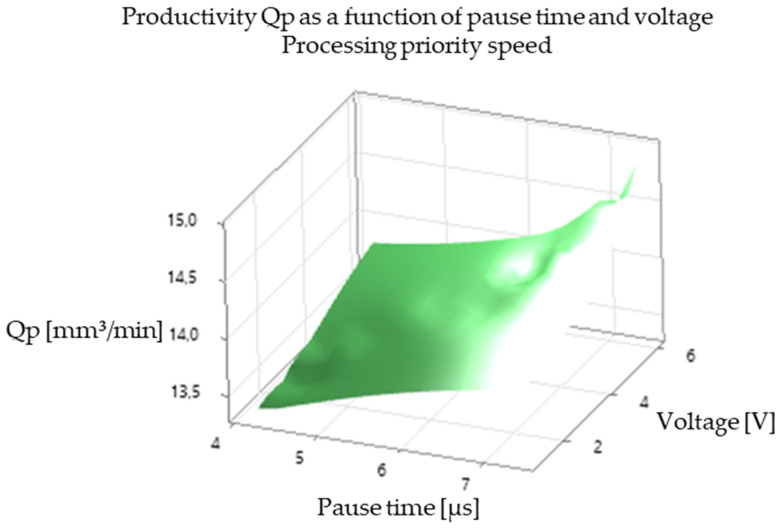
Productivity vs. break time and tension. Processing priority wear rate.

**Figure 13 polymers-16-03019-f013:**
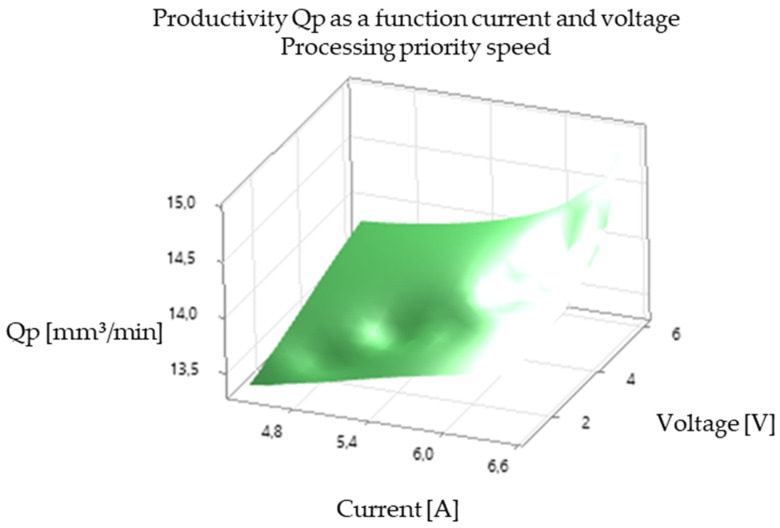
Productivity vs. current and voltage. Processing priority wear rate.

**Table 1 polymers-16-03019-t001:** Input data—Variant 1—Standard processing priority.

Duration: 129 min	Standard Priority Input Data				Objective Function
Eroded Volume: 1074.6 mm^3^	$X_{1}$	$X_{2}$	$X_{3}$	$X_{4}$	$Y_{1}$
No.	Impulse Time $t_{i}$ [μs]	Break Time $t_{p}$ [μs]	Current Intensity I [A]	Tension U [V]	Productivity $Q_{p}$ [mm^3^/min]
1	8.4	7.5	6.5	5	8.36
2	8.3	7.4	6.4	5	8.36
3	8.2	7.3	6.3	5	8.34
4	8.1	7.2	6.2	4	8.32
5	8.0	7.1	6.1	4	8.30
6	7.9	6.9	5.9	4	8.27
7	7.7	6.7	5.7	3	8.26
8	7.5	6.6	5.2	3	8.20
9	7.2	6.2	4.8	2	8.15
10	6.6	5.7	4.4	1	8.04

**Table 2 polymers-16-03019-t002:** Input data—Variant 2—Priority of machining with low electrode wear rate.

Duration: 152 min	Input Data Priority Wear Rate				Objective Function
Eroded Volume: 1080 mm^3^	$X_{1}$	$X_{2}$	$X_{3}$	$X_{4}$	$Y_{1}$
No.	Impulse Time $t_{i}$ [μs]	Break Time $t_{p}$ [μs]	Current Intensity I [A]	Tension U [V]	Productivity $Q_{p}$ [mm^3^/min]
1	5.9	7.4	6.5	2.2	7.53
2	5.7	7.1	6.4	2.1	7.51
3	5.5	6.8	6.3	1.9	7.45
4	5.3	6.3	6.2	1.8	7.31
5	5.2	6.2	6.1	1.7	7.25
6	5.1	5.9	5.8	1.5	7.24
7	4.9	5.6	5.5	1.4	7.22
8	4.8	5.3	5.2	1.3	7.13
9	4.6	5.0	4.9	0.9	7.09
10	4.4	4.9	4.4	0.4	7.06

**Table 3 polymers-16-03019-t003:** Input—Variant 3—Processing speed priority.

Duration: 80 min	Speed Priority Input Data				Objective Function
Eroded Volume: 1069 mm^3^	$X_{1}$	$X_{2}$	$X_{3}$	$X_{4}$	$Y_{1}$
No.	Impulse Time $t_{i}$ [μs]	Break Time $t_{p}$ [μs]	Current Intensity I [A]	Tension U [V]	Productivity $Q_{p}\;[{\mathrm{mm}}^{3}/\min]$
1	5.9	7.4	6.5	6	14.86
2	5.8	7.2	6.4	6	14.51
3	5.5	6.7	6.3	5	14.23
4	5.2	6.4	6.2	5	13.88
5	4.9	5.9	6.1	4	13.82
6	4.6	5.5	5.9	4	13.75
7	4.3	4.6	5.5	3	13.62
8	4.1	4.4	5.2	3	13.43
9	3.9	4.3	4.8	2	13.37
10	3.6	4.2	4.4	1	13.36

**Table 4 polymers-16-03019-t004:** Important parameters in the electrical erosion process.

	Ra [µ m]	Qp Spec [mm^3^/min]	Θ [%]
Roughing	>3	4.5–9.0	0.2–0.01
Finishing	0.8–3	0.3–4.5	2.4–0.2
Rectification	0.5–0.8	<0.3	>15–2.4

**Table 5 polymers-16-03019-t005:** Optimal EDM parameter settings for each processing strategy.

Processing Strategy	Discharge Current (I)	Pulse Time (Ton)	Pause Time (Toff)	Voltage (V)
Standard	6.5	8.4	6.2	5.5
Low Electrode Wear	6.5	5.9	5	2.2
High-Speed	6.5	5.9	4.6	6

**Table 6 polymers-16-03019-t006:** Comparison of MRR with initial and optimized parameter settings.

Processing Strategy	Initial MRR (mm^3^/min)	Optimized MRR (mm^3^/min)	Improvement (%)
Standard	8.04	8.36	+3.98
Low Electrode Wear	7.06	7.53	+6.66
High-Speed	13.36	14.86	+11.22

## Data Availability

Data are contained within the article.

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
