# Peer review of "Enhancing EDM Productivity for Plastic Injection Mold Manufacturing: An Experimental Optimization Study"

_polymers, 2024, doi:10.3390/polym16213019_

Round 1

Reviewer 1 Report

Comments and Suggestions for Authors

Dear Authors,

In general, the EDM process is very popular in manufacturing, and your topic aligns well with this strength. To evaluate your research work, there are some issues that you need to clarify:

  1. Informally, you should use the working email (if possible), so please check the email.
  2. Line 104: How did other research measure productivity?
  3. At the end of Part 1, the reason for researching this topic must be shown. (Is there any research done on these parameters?)
  4. Lines 119–123: Please discuss the reason for selecting these parameters.
  5. Lines 119–123: Please describe and discuss clearly the influence of each parameter on electrode wear, surface quality, etc.
  6. Tables 1, 2, 3: How did you select these values for this research? In addition, please describe and discuss clearly Tables 1, 2, and 3.
  7. At the end of Part 3, please add the method for measuring productivity as well as other quality issues.
  8. The title of “Part 4.2”: Impact of processing strategy on what?
  9. Part 4.4: In my opinion, this is not an optimization process. This part just describes or concludes the influence of EDM parameters on productivity. So, please check the target of this manuscript, as well as the title of this paper.

Sincerely yours,

Reviewer 2 Report

Comments and Suggestions for Authors

1. Overall, the manuscript is lack of quantitative data presentation and high-quality discussion about the experiments performed. The figures (Figure 5-13) were not fully discussed. The trend of the EDM factors are not presented in a technical format.

2. I would suggest authors to include more quantitative information of the research (geometric accuracy) in the abstract.

3. Section 2: the authors should include information about characterization methods and equipment used for electrode wear and surface finish.

4. The caption of Figure 5, 8, and 11 are the same. The differences should be presented in the captions.

5. Section 4: The data can be better presented using statistical analysis methods, such as factorial DOE. I believe it will provide more insight into the trends of factors on the responses.

6. Section 4.3: The wear and surface finish data should be quantitatively presented in the manuscript.

Round 2

Reviewer 1 Report

Comments and Suggestions for Authors

Dear Authors,

The last version is good for publishing. However, you should reduce the “Conclusion” part become shorter.

Sincerely yours,

Reviewer 2 Report

Comments and Suggestions for Authors

The revised manuscript is improved, and proper for publication. Thanks!
